# Strategies for Safe Transurethral Injections of Botulinum Toxin into the Bladder Wall

**DOI:** 10.3390/toxins16070299

**Published:** 2024-06-30

**Authors:** Matthias Oelke

**Affiliations:** 1Hannover Medical School, Siedlerweg 10, 48599 Gronau, Germany; dr.moelke@t-online.de; Tel.: +31-6-29-74-15-52; 2Kantonsspital Frauenfeld, Spital Thurgau AG, Waldeggstr. 8A, 8500 Frauenfeld, Switzerland

**Keywords:** botulinum toxin, overactive bladder, detrusor overactivity, neurogenic bladder dysfunction, detrusor wall thickness, injection

## Abstract

**Introduction:** Transurethral injections into the bladder wall with botulinum toxin are an established treatment for refractory overactive bladder or detrusor overactivity. With the current injection technique, an average of approx. 18% and up to 40% of botulinum toxin is injected next to the bladder wall, potentially causing reduced efficacy or non-response. The article aims to evaluate the reasons for incorrect injections and propose strategies for complete delivery of the entire botulinum toxin fluid into the bladder wall. **Material and Methods:** Unstructured literature search and narrative review of the literature. **Results:** Incorrect injection of botulinum toxin fluid next to the bladder wall is caused by pushing the injection needle too deep and through the bladder wall. Bladder wall thickness decreases with increasing bladder filling and has a thickness of less than 2 mm beyond 100 mL in healthy individuals. Ultrasound imaging of the bladder wall before botulinum toxin injection can verify bladder wall thickness in individual patients. Patient movements during the injection therapy increase the chance of incorrect placement of the needle tip. **Conclusions:** Based on the literature search, it is helpful and recommended to (1) perform pretreatment ultrasound imaging of the bladder to estimate bladder wall thickness and to adjust the injection depth accordingly, (2) fill the bladder as low as possible, ideally below 100 mL, (3) use short needles, ideally 2 mm, and (4) provide sufficient anesthesia and pain management to avoid patient movements during the injection therapy.

## 1. Introduction

Transurethral injections of botulinum toxin into the bladder wall are a recommended and well-accepted treatment option for female or male patients with treatment-refractory overactive bladder or (neurogenic or non-neurogenic) detrusor overactivity with or without urinary urgency incontinence [1,2,3,4]. Therefore, transurethral botulinum toxin injections should be considered when previous treatments have failed, such as conservative treatment (e.g., behavioral modifications, lifestyle advice), pelvic floor muscle exercise with or without biofeedback, drug treatment with muscarinic receptor antagonists and/or β_3_-adrenoceptor agonists, and electrostimulation. Although not licensed for this indication, botulinum toxin may also be applied as a fourth-line treatment for interstitial cystitis/bladder pain syndrome, as recommended by the AUA guidelines [5,6].

Botulinum toxin is a bacterial neurotoxin with 1296 amino acids and a molecular weight of 146 kDa produced by several *clostridium* species, of which *clostridium botulinum*, *clostridium butyricum*, *clostridium baratii,* or *clostridium argentinense* are the most important [7]. Of the seven known subtypes of botulinum toxin, only types A and B are commercially available, of which onabotulinum toxin A since 2013 (Botox^®^; Allergan Pharmaceuticals, Dublin, Ireland) and abobotulinum toxin A since 2022 (Dysport^®^; Ipsen, Boulogne-Billancourt, France) are licensed for the treatment of overactive bladder or (neurogenic) detrusor overactivity. In comparison, Botox^®^ has a total molecular weight of approx. 900 kDa (146 kDa neurotoxin + approx. 750 kDa protein chains) and Dysport^®^ has a molecular weight of approx. 500 kDa (146 kDa neurotoxin + approx. 350 kDa protein chains) [8]. Botulinum toxin from both companies is stored as white crystalline powder (lyophilizates) in vacuum-sealed glass bottles and must be dissolved in sterile saline solution before use. Because of the high molecular weight of both botulinum toxin formulations and the impermeable urothelial layer, botulinum toxin must be injected into the bladder wall to reach the target tissue, i.e., the parasympathetic nerve terminals of the detrusor and/or submucosa, thereby inhibiting acetylcholine release into the synaptic cleft [9] and, finally, decreasing bladder sensations as well as detrusor muscle contractions [10]. Submucosal injections of botulinum toxin have been shown to be as effective as intradetrusor injections [11]. Consequently, the botulinum toxin solution must be injected into the submucosal layer and/or detrusor, whereas injections outside these tissue layers (perivesical space) do not reach the parasympathetic nerve terminals and, therefore, do not show clinical effects.

Mehnert et al. noted that a standardization of the injection technique is still missing and, with the help of magnetic resonance imaging (MRI) and contrast media, demonstrated that an average of approx. 18% of the injected botulinum toxin solution was eventually injected into the perivesical fatty tissue (average of 53 U of the 300 U Botox^®^) [12]. The authors investigated six adult patients aged 18 to 82 years with neurogenic detrusor overactivity incontinence due to spinal cord injury at levels Th6 to Th11. They used rigid cystoscopes and 22G needles with a length of 8 mm, which were retracted by approx. half of the length (i.e., 4 mm) after pricking the bladder wall at the dome and base. Immediately after the injections, MRI was performed, the contrast media inside and outside the bladder wall visualized, and the volume of contrast media in both locations calculated. In men, 8.2–15.9 cm^3^ (14.6%) of the injected contrast media was detected lateral of bladder walls (i.e., 44 U Botox^®^) and, in women, 2.5–14.4 cm^3^ (20.8%; i.e., 62 U Botox^®^). One female patient had spastic limb contractions during the operation, and the injection procedure with withdrawing the needle by approx. half of the length for bladder wall injections was more difficult; consequently, 40% of the total injected contrast media solution was located lateral of the bladder walls (120 U Botox^®^). The majority of perivesical contrast media was seen lateral to the bladder dome and, in one patient, beyond the bladder base. Unfortunately, the authors did not report on the bladder filling volume during the botulinum toxin injections.

Based on the results of this study, it becomes evident that perivesical injections of botulinum toxin were caused by pushing the tip of the injection needle through the bladder wall. Although the needle was retracted by approx. half of its length (4 mm), this maneuver did not prevent false injections. The question arises: how deep can the injection needle be inserted into the bladder wall to safely deliver the total botulinum toxin fluid to the target location? This article deals with theoretical and practical considerations of the injection technique for safe delivery of (nearly) 100% of the botulinum toxin dose into the bladder wall.

## 2. Bladder Wall Thickness in Relation to Bladder Filling Volume

The bladder wall can be imaged well with high-frequency ultrasound probes (e.g., 7.5 MHz) by the suprapubic or transrectal route in men as well as the suprapubic, transrectal, transvaginal, or transperineal route in women [13,14]. The technique of ultrasound detrusor wall thickness (DWT) measurements has been described earlier [15]. The mucosa and adventitia of the bladder wall appear hyperechogenic (white), while the detrusor in between these two layers appears hypoechogenic (black, Figure 1) [14,16,17]. Initial studies in healthy, young adult women and men focused on the relationship between bladder/detrusor wall thickness and bladder filling volume and demonstrated a decreasing thickness with increasing bladder filling [17]. In this study, detrusor wall thickness (DWT) was measured every 50 mL until 300 mL of bladder filling volume and, afterwards, every 100 mL until maximum bladder capacity. The study showed in both women and men a rapidly decreasing DWT during the bladder filling until approx. 100 mL and a slower decrease until bladder capacity (Figure 2). DWT was measured at approx. 2.5–4.5 mm at low bladder filling and reached an average DWT of 1.2 mm (women) to 1.4 mm (men) at capacity.

This study on the relationship between DWT and bladder filling shows that it is crucial to know and control the bladder filling volume during botulinum toxin injections. Consequently, botulinum toxin fluid is correctly and completely delivered into the bladder wall when the physician uses a short needle and the bladder is only filled with a low volume (e.g., 50–100 mL), whereas the needle with the same length is pushed through the entire bladder wall and botulinum toxin fluid is injected outside the bladder wall when the bladder is filled with a higher volume (e.g., 300–400 mL).

## 3. Adjustments to the Injection Technique

The author of this current article modifies his injection technique by pretreatment ultrasound imaging of the bladder wall in all patients because bladder wall thickness, compared to healthy individuals, can be increased in patients with bladder outlet obstruction [15,18] (Figure 3) as well as in some patients with neurogenic bladder dysfunction (e.g., low compliance due to myelomeningocele/spina bifida or paraplegia) [19,20] or decreased in patients with detrusor underactivity [21]. Routine ultrasound imaging of the bladder wall provides useful information about how deep the needle can be inserted into the bladder wall for safe delivery of the entire botulinum toxin fluid, especially in patients with suspected thinner bladder walls, e.g., younger patients (children or teenagers), women, and those with suspicion of detrusor underactivity.

The injection technique must be further modified when using rigid cystoscopes because the bladder dome (cranial part of the bladder) is usually penetrated by the needle in a perpendicular (orthograde) fashion, whereas the lateral bladder walls and the bladder base are usually penetrated in a tangential direction, the latter locations providing a longer distance of the injection route (Figure 4). Therefore, the author of this current article fills the bladder only with low volume (<100 mL), uses 4 mm needles that are retracted by approx. half of their length (2 mm) after pricking the bladder wall and before injection of the botulinum toxin fluid, and prefers tangential injections wherever possible.

## 4. Discussion

The authors of the MRI-based study demonstrated that an average of 18% (and up to 40%) of the botulinum injection fluid was delivered next (laterally) to the bladder after pricking the bladder wall and inserting the needle by approx. 4 mm [12]. Their study was conducted in patients with paraplegia who usually have thickened bladder walls due to the underlying disease [19,20]. It can be hypothesized that a larger and more substantial amount of botulinum toxin fluid would have been injected beyond the bladder wall when using the identical injection technique and treating patients without thickened bladder walls (e.g., women with non-neurogenic overactive bladder). This may be one reason why transurethral botulinum toxin treatment remains ineffective in some patients.

For safe and complete delivery of the botulinum toxin fluid into the bladder wall, it is important to minimize the bladder filling, which, however, should be high enough to see all parts of the bladder during the procedure. This situation is usually achieved when the bladder has been filled with 50–100 mL. The penetration depth of the needle should also be adjusted to the location of the bladder wall injections (dome vs. lateral walls). The shorter the needle or the lower the penetration depth of the needle, the higher the chance of correct and complete injections of botulinum toxin fluid. The industry has lately provided short needles for this purpose (e.g., injeTAK^®^ cystoscopy needles with an adjustable injection depth between 2 and 5 mm with 1 mm increments; Laborie, Portsmouth, NH, USA). These needles must not be retracted after pricking of the bladder wall, especially when bladder wall thickness is known in the individual patient by ultrasound investigation before the procedure. Although superficial injections are more likely with these needles, previous studies and a meta-analysis have shown similar efficacy of submucosal vs. intradetrusor injections [11]. Submucosal injections can be visualized by bulking of the mucosa into the bladder lumen at the injection site, whereas this bulking is missing with intradetrusor (or perivesical) injections.

Adequate anesthesia for painless injections is another factor contributing to the correct placement of the botulinum toxin dose into the bladder wall. It was demonstrated in the study by Mehnert et al. that patient movements during pricking of the bladder wall and/or retracting the needle before the injections can result in perivesical placement of the botulinum toxin solution of up to 40% of the total injected dose [12]. Sufficient anesthesia of the bladder can be achieved by transurethral instillation of (alkalized) lidocaine or related anesthetics, regional, or general anesthesia. Local anesthesia has the advantage of avoiding the presence of anesthesiologists, intra- and postoperative screening of the patient’s vital parameters, and anesthesia-related risks. The author of this current article has been using local anesthesia for the last 4 years in approx. 300 patients without or only minimal pain (visual analog pain 0–2 on a Likert scale reaching from 0 to 10). He uses transurethral bladder installations with 50 mL lidocaine 2% in combination with 50 mL sodium bicarbonate 8.4% at a temperature of approx. 5 °C (refrigerator) 20 to 30 min before the botulinum toxin injection treatment. Pereira e Silva et al. demonstrated in a double-blind, randomized, controlled study that lidocaine and sodium bicarbonate together significantly decrease pain scores better than lidocaine alone [22]. Immediately before cystoscopy and bladder wall injections, the author of this current article additionally injects half (women) or one syringe (men) with 11 mL lidocaine-chlorhexidine gel (InstillaGel^®^; Farco Pharma, Cologne, Germany) at a temperature of approx. 5 °C (refrigerator) into the urethra for lubrication, disinfection, and anesthesia. Electromotive drug administration (EMDA) with lidocaine 4% has also been shown to significantly decrease pain during botulinum toxin injections compared to lidocaine instillations alone [23]. In rare cases, reduction in the injection sites (without decreasing the total botulinum toxin dose) [24,25] or oral phenazopyridine, which is excreted via the kidneys and has a local anesthetic effect on the urinary tract, can also significantly reduce pain during botulinum toxin treatment [26].

Intravesical instillation of liposome-encapsulated botulinum toxin A (lipo-botulinum toxin) could overcome the issues of perivesical injections, bladder filling, and pain management in the future. Liposomes are small lipophile two-layer phospholipid vesicles that can fuse with cell membranes, pass the usually impermeable urothelial layer, and release the content of the aqueous core with botulinum toxin for safe delivery of the drug into the target tissue in the submucosa or detrusor [27,28]. Despite promising pilot studies, lipo-botulinum toxin has not yet become commercially available.

## 5. Conclusions

Physicians aim to inject the intended botulinum toxin dose completely into the bladder wall, thereby avoiding perivesical injections and reducing the clinical effects. A summary of the methods supporting complete injection of the botulinum toxin fluid is visualized in Figure 5. Preoperative visualization of the bladder wall helps judge bladder wall thickness in individual patients. Because bladder wall thickness decreases with increasing bladder filling volume, it is necessary to fill the bladder as low as possible but high enough to see all parts of the bladder. This situation is usually achieved when the bladder has been filled with a physiological saline solution of <100 mL. In addition, physicians should use short needles (e.g., 2 mm) for safe injections of the botulinum toxin fluid into the submucosal layer of the bladder and/or detrusor. Adequate pain management during the injection therapy (e.g., lidocaine-sodium bicarbonate instillations 20–30 min before) further contributes to the safe and predictable application of botulinum toxin. The proposed strategies are important for predictable, reliable, and reproducible clinical results, especially during re-injections and up- or down-titration of the botulinum toxin dose in patients with too low or too large effects during the first botulinum toxin treatment(s).

## Figures and Tables

**Figure 1 toxins-16-00299-f001:**
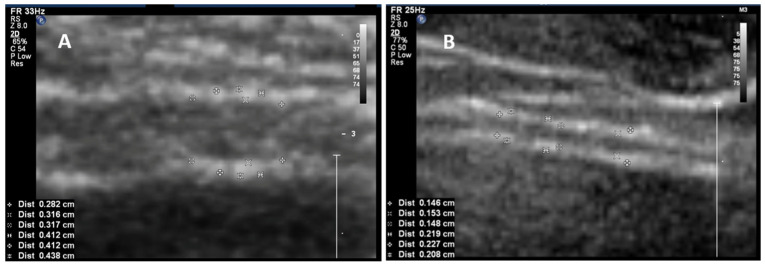
Imaging of the anterior bladder wall with a suprapubically positioned ultrasound probe in a patient aged 46 years with overactive bladder/detrusor overactivity, both images 6-times enlarged. The detrusor appears hypoechogenic (black) and is sandwiched between the mucosa and adventitia, which both appear hyperechogenic (white). The detrusor (upper three measurements, i.e., detrusor wall thickness) and the entire bladder wall (lower three measurements, i.e., bladder wall thickness) were measured. (**A**) Bladder wall of the bladder filled with 50 mL and (**B**) with 250 mL (source: private image collection M. Oelke).

**Figure 2 toxins-16-00299-f002:**
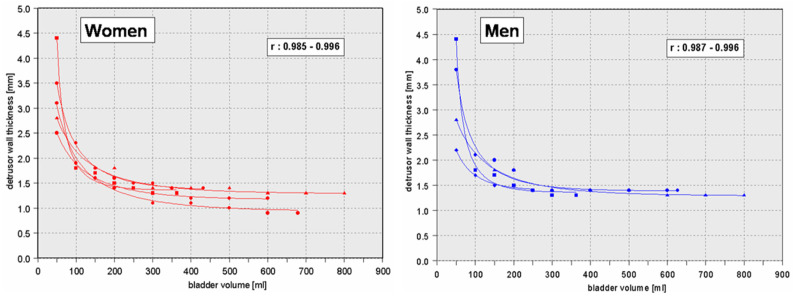
Relationship between bladder filling (*x*-axis) and detrusor wall thickness (*y*-axis) in four different healthy women (**left**) and four healthy men (**right**). Measurements of the same person at different bladder fillings were connected with lines. Detrusor wall thickness rapidly decreases during the first 100–150 mL of bladder filling but only decreases slightly thereafter [17].

**Figure 3 toxins-16-00299-f003:**
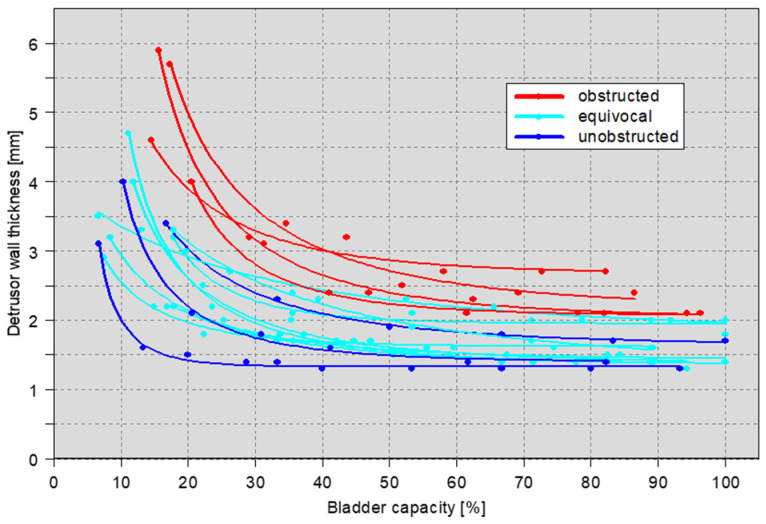
Relationship between bladder capacity (*x*-axis) and detrusor wall thickness (*y*-axis) in men with benign prostatic hyperplasia: without (dark blue), equivocal (light blue), and with bladder outlet obstruction (red). In contrast to Figure 2, bladder filling (ml) was changed to % bladder capacity to adjust for the differences in bladder capacity in the individual patients. Measurements of the same person at different bladder fillings were connected with lines. Like healthy individuals, detrusor wall thickness rapidly decreases during the first phase of bladder filling but only slightly thereafter. The extent of bladder wall hypertrophy (detrusor wall thickness) is dependent on the degree of bladder outlet obstruction (source: private image collection, M. Oelke).

**Figure 4 toxins-16-00299-f004:**
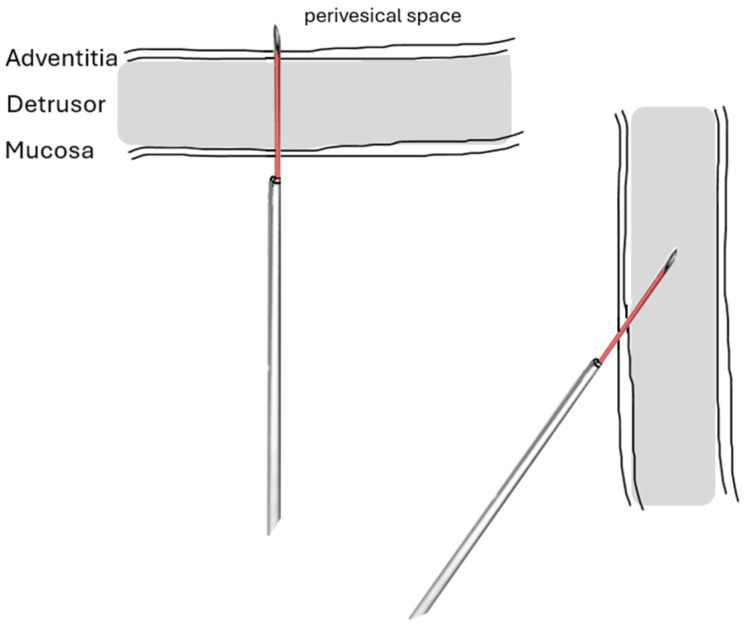
Illustration of perpendicular (e.g., at the bladder dome) and tangential insertion of the injection needle (e.g., at the lateral bladder wall or bladder base). In this figure, a needle length of 4 mm was chosen. Tangential insertion provides a longer distance within the bladder wall and, therefore, a higher chance to inject botulinum toxin at the correct location (source: private image collection, M. Oelke).

**Figure 5 toxins-16-00299-f005:**
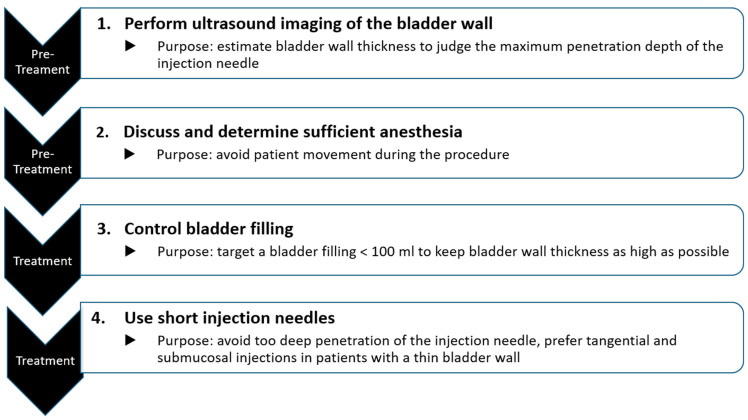
Summary of the four methods to minimize the chance of perivesical injections of botulinum toxin fluid into the bladder wall.

## Data Availability

Not applicable.

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
