# Peer review of "Strategies for Safe Transurethral Injections of Botulinum Toxin into the Bladder Wall"

_toxins, 2024, doi:10.3390/toxins16070299_

Round 1
Reviewer 1 Report
Comments and Suggestions for Authors
This manuscript reported the injection technique of intravesical botulinum toxin A (BoNT-A) treatment for patients with overactive bladder/detrusor overactivity. There are several drawbacks in this article.
1) Is this a review article or investigative article? From the structure of manuscript, it appears to be a review article.
2) The title of this article is not appropriate. Improving injection technique can only ensure injecting most of the BoNT-A solution into the detrusor wall, but not predict the successful treatment for BoNT-A injection.
3) Currently, BoNT-A has been applied not only OAB but also in patients with neurogenic detrusor overactivity (NDO) and interstitial cystitis (IC). The injection technique not only the depth and angle of injection, but also the injecting sites in patients with different bladder conditions such as OAB with low contractility, male patients with bladder outlet obstruction. The authors should mention more for the readers to understand and choose the best injection technique and program for a safe and successful treatment outcome (see reference: Jiang YH, Jhang JF, Kuo HC. Tzu Chi Med J 2023; 35: 31-7)
4) Bladder wall sonography is not available for most clinic. The bladder wall thickness is not an essential part for a successful BoNT-A injection. However, a suburothelial injection can provide a way to increase the percentage of BoNT-A solution injecting into the bladder wall. There have been many literature reports, please also discuss this point.
Author Response
Please find it in the attachment.

Reviewer 2 Report
Comments and Suggestions for Authors
The paper addresses a significant issue in the treatment of overactive bladder with botulinum toxin injections. It proposes a well-founded technique to improve injection accuracy, supported by existing literature. The paper is well-structured and provides practical recommendations for clinical practice. However, it could benefit from a more detailed methodology section and a discussion of potential implementation challenges.
The introduction could be more concise. You can conclude at line 54 adding the aim of the paper. It could be added another section to better analyze the injection technique
The authors should include detailed methodology for the literature search to allow replication and validation of the study.
Consider adding a flowchart summarizing the steps of the proposed technique.
The results could be strengthened by providing a summary table of the key findings from the referenced studies.
Discuss any limitations of the proposed technique based on the reviewed literature.
Mention any future research directions or ongoing studies that could further validate the proposed technique.
Minor corrections:
Line 33: add reference
Line 40: FDA approvations?
Line 75: re-add the reference for clarity
Line126: When the author uses the phrase "The author of this article", it is not clear whether the authors are referring to the present article or citing one.
Author Response
Please find it in the attachment.

Round 2
Reviewer 1 Report
Comments and Suggestions for Authors
The authors have adequately respond to my comments and the revised manuscript is currently acceptable for publication in this journal.